# Optimal Histopathological Magnification Factors for Deep Learning-Based Breast Cancer Prediction

**Abduladhim Ashtaiwi** 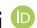

College of Engineering and Technology, American University of the Middle East, Egaila 15453, Kuwait; abduladhim.ashtaiwi@aum.edu.kw; Tel.: +965-69336918

**Abstract:** Pathologists use histopathology to examine tissues or cells under a microscope to compare healthy and abnormal tissue structures. Differentiating benign from malignant tumors is the most critical aspect of cancer histopathology. Pathologists use a range of magnification factors, including 40x, 100x, 200x, and 400x, to identify abnormal tissue structures. It is a painful process because specialists must spend much time sitting and gazing into the microscope lenses. Hence, pathologists are more likely to make errors due to being overworked or fatigued. Automating cancer detection in histopathology is the best way to mitigate humans' erroneous diagnostics. Multiple approaches in the literature suggest methods to automate the detection of breast cancer based on the use of histopathological images. This work performs a comprehensive analysis to identify which magnification factors, 40x, 100x, 200x, and 400x, induce higher prediction accuracy. This study found that training Convolutional Neural Networks (CNNs) on 200x and 400x magnification factors increased the prediction accuracy compared to training on 40x and 100x. More specifically, this study finds that the CNN model performs better when trained on 200x than on 400x.

**Keywords:** breast cancer detection; histopathological image; machine learning; deep learning; convolutional neural networks; feature extraction; image preprocessing

## 1. Introduction

The most common histopathological dye is hematoxylin and eosin (H&E). This stain colors cell nuclei blue and cytoplasm pink. Cancer cells typically appear abnormal when stained with H&E; the cell structure's appearance helps the pathologist distinguish them from normal cells. A biopsy or surgical excision is the most common source of a tissue specimen. A needle is inserted into the tumor during a biopsy, and cells are extracted for examination. Surgical excision is when the entire tumor or lesion is surgically removed and sent to a pathologist for analysis. Pathology microscopes are equipped with different magnification capabilities: low-power magnification ranges from 2x to 10x; medium-power magnification ranges from 10x to 40x; and high-power magnification ranges from 40x to 400x, depending on the pathologist's needs. By examining different magnification factors, the histopathologist can discover unique features of the deforming cells, which helps diagnose whether the cells are malignant or benign. Microscopic tissue analysis assists in determining the disease stage and guiding treatment decisions. While histopathology is critical, it can also be prone to diagnostic error because histopathologists often look at slides for a long time and become tired. However, errors can still occur with the utmost care and precision. The most recent study on histopathologist error statistics found that while pathologists make an estimated 1 in 10,000 mistakes when diagnosing actual cancer cells, errors rise to 1 in 3 when diagnosing earlier stages of cancer cells due to their indistinct cancer cell features (Valenstein et al. [1]). The National Academy of Medicine (NAM) [2] found that pathology mistakes account for up to 10 percent of hospital adverse patient events. Many factors can contribute to pathology errors, including inadequate training, poor communication among healthcare providers, and faulty equipment. Many

experts believe that the number of actual errors committed is likely far higher than reported due to underreporting by physicians and patients. With continued research into new techniques and technologies, it is believed that Artificial Intelligence (AI) and Deep Learning (DL) are promising technologies to help pathologists reduce diagnostic errors. Convolution Neural Networks (CNNs) are a type of Machine Learning (ML) that uses neural networks to comprehend how to identify patterns in images. CNNs have made it possible to detect cell cancer biomarkers more accurately than ever. For example, one study showed that CNN-based algorithms could correctly pinpoint breast cancer metastases with a precision of 90% and above most of the time—significantly outperforming human pathologists. Many ideas in the literature are based on using ML and histopathological images, combined with various data preprocessing techniques, to predict breast cancer. Authors in Anjum et al. [3], Gupta et al. [4], Yadav et al. [5], Anwar et al. [6], and Gultekin et al. [7] have applied Support Vector Machine (SVM), while the works by Gultekin et al. [7] and Vo-Le et al. [8] utilize Logistic Regression (LR). The studies by Anwar et al. [6], Hohn et al. [9], Dabeer et al. [10], Wadhwa et al. [11], Khuriwal et al. [12], Qi et al. [13], and El-Agouri et al. [14] predict breast cancer using CNNs. The study by Da et al. [15] utilizes CNNs to predict a malignant tumor of the digestive system. Residual Neural Networks (RNNs) are implemented in the work of Chatterjee et al. [16], the Random Forest (RF) algorithm is applied in Carvalho et al. [17] and Anwar et al. [6]. Similarly, fuzzy classification is used in Qidwai et al. [18]; K-means in Yadav et al. [5]; SENet DL in Chen et al. [19]; the Naive Bayes (NB) and Decision Tree (DT) in Vo-Le et al. [8]. The algorithms above are combined with other data preprocessing feature extraction techniques such as DenseNet, Histograms of Oriented Gradient (HOG), Wavelet Packet Decomposition (WPD), ResNet, Principal Component Analysis (PCA), Region of Interest (RoI) extraction, VGG-16, and many others. Another approach was done by Foroughi et al. [20] to study the biological features interpretation of histopathology to enhance the prediction accuracy of CNNs. As stated in previous works, they utilize histopathological images and ML to predict cancerous and non-cancerous tissues. Many studies have achieved higher accuracy, making breast cancer prediction using histopathological images and ML a promising practice. Next, the research question analyzed in this paper is explained.

*Problem Statement*

Different magnification factors, such as low, medium, and high magnification power, are a crucial part of the histopathology study. Low-power microscopes offer a wide field of view and can be used to examine significant areas of tissue quickly. However, they provide little detail and cannot be used to identify individual cells or structures. Medium-power microscopes offer more detail than low-power microscopes but cannot cover as large an area as high-power microscopes. They are ideal for identifying small structures within tissue samples. High-power microscopes offer the most significant level of detail but can only examine minimal areas at a time. High-power microscopes are ideal for identifying individual cells or structures within tissue samples. Pathologists benefit from different magnification factors to see different cell representations and spot abnormalities. The information gleaned from the various magnification levels is typically used by pathologists to develop a general understanding of the health status of the histopathological image under study. Similar to a pathologist, this study examines how well the learning algorithm apprehends information from the various magnification levels of histopathological images. Using the most appropriate magnification factor that achieves the highest learning rate is crucial to improving the model's accuracy. Inappropriate magnification factors may be considered noisy, which degrades the model's accuracy. Hence, the questions this study endeavors to address are: Do machine learning algorithms also benefit from the existence of different magnification factors? Which magnification factor represents high-quality data, and which one represents noisy data? The above questions are answered utilizing the BreakHis [21] dataset, which contains four magnification factors, i.e., 40x, 100x, 200x and 400x.

The remainder of this work is organized as follows: Section 2 reviews related works in the literature and contrasts their conclusions with the findings of this work. The background information required to understand the context of this work is explained in Section 3. Section 4 describes the methodology employed in this research paper in detail. Detailed result analysis and an explanation of this work's findings are provided in Section 5. Section 6 reweighs the conclusions of this work to those presented in the literature review. Finally, closing remarks are given in Section 7.

## 2. Literature Review

Numerous methods have been proposed in the literature that aim to predict breast cancer using ML, histopathological images, and various data preprocessing approaches. Some researchers believe data preprocessing techniques are essential for accurately predicting disease states from medical images. Others believe ML algorithms can be used effectively without any preprocessing steps. Still, others think combining data preprocessing and ML may produce the most accurate predictions possible. All of the papers chosen for this study's literature review use histopathological images and ML algorithms to identify cancerous tissue, mainly in the breast. The work by Chatterjee et al. [16] detects Invasive Ductal Carcinoma (IDC) (a type of breast cancer) using Residual Convolution Networks (RCNs) to classify the IDC-affected histopathological images from the normal images. First, the microscopic RGB images are converted into a seven-channel image matrix, then fed to RCNs. The work claims that the proposed model produces an accuracy of over 90%. The study by Anjum et al. [3] emphasizes that feature extraction from images plays a prominent role in image processing. Their study applies a combination of histograms of oriented gradient and Canny Edge Detection (CED) techniques for extracting features. Then they use PCA to reduce the dimensionality of the extracted features. PCA output is input for SVM and LR. The experiment shows 94% correct detection of malignant patients. The authors in Hohn et al. [9] investigated whether combining histopathological images with commonly available patient data (such as age, sex, and anatomical site of the lesion) could increase the performance compared with CNNs alone, referred to as standard CNNs. Their results showed that standard CNNs achieve better accuracy in most cases than patient data integration with the image. The authors in Carvalho et al. [17] introduce an approach to quantify and classify breast tissue samples based on features extracted from the intensity histogram, co-occurrence matrix, Shannon, Renyi, Tsallis, and Kapoor entropies. The obtained feature vector is used as input to the RF and Sequential Minimal Optimization (SMO) algorithms. The study by Carvalho et al. [17] claims to achieve significant results in the Area Under the Curve (AUC) performance measurements. The authors in Gupta et al. [4] employ two ML algorithms for comparative analysis: SVM and LR. The model is trained separately for various image magnification factors, i.e., 40x, 100x, 200x and 400x. The findings of Gupta et al. [4] demonstrate that the ResNet50 has achieved maximum accuracy for LR compared to SVM in magnification factor. In addition, results show that the performance of CNNs+LR is slightly better than CNNs+SVM for classifying benign and malignant classes. The work by Dabeer et al. [10] utilizes CNNs to extract the features and create a model. The Fuzzy Classifier is utilized in Qidwai et al. [18] to quantify and classify cancerous cells based on color. The color analysis is based on various colors, such as Hue Saturation Value (HSV), rather than specific color values. Their findings can also be used in the morphological processing of the classified binary image to locate, count, and confirm particular kinds of regions and tissues in the biopsy sample. The study by Wadhwa et al. [11] exploits DenseNet-201 as a feature extraction method. Their study achieves above 90% accuracy while the precision and recall, ML performance metrics, are 0.90 and 0.99, respectively; the F1-score is reported to be 89%. The authors in Yadav et al. [5] apply two procedures: first, they use ML algorithms for image classification; second, they use segmentation algorithms for detecting tumorous cells; and afterward, they use SVM and CNNs algorithms for tumorous classification. The classifiers are examined based on sensitivity, specificity, accuracy, precision, and F1-score parameters. The resulting images

are further used as an input for image segmentation utilizing Genetic Algorithms (GA) and K-Means. The proposed methodology in Anwar et al. [6] consists of four stages: image preprocessing; feature extraction (using ResNet); feature reduction (using PCA); and last classification using SVM, RL, and Quadratic Discriminate Analysis (QDA). The authors in Hirra et al. [22] employ Deep Belief Networks (DBNs) for histopathological image classification. Features are extracted through an unsupervised pre-training and supervised fine-tuning phase. The features extracted from the patches are fed to the model as input. The model presents the result as a probability matrix, either a positive sample (cancer) or a negative sample (background). The authors in Vo-Le et al. [8] employ a combined feature extraction algorithm: VGG-16, GoogLeNet, or ResNet-50. For the classification, they use NB, DT. The findings demonstrate that using VGG-16 as feature extraction helps the model achieve slightly higher accuracy. The study by Mohalder et al. [23] predicted lung cancer by using CatBoost, DT, LR, and Linear Discriminant Analysis (LDA); Catboost achieved the highest prediction accuracy. The authors of Khuriwal et al. [12], Haija et al. [24], and Jannesari et al. [25] utilize the CNNs algorithm for cancer classification. However, they apply different feature extraction techniques; the authors in Khuriwal et al. [12] use the Entropy function, while the authors of Haija et al. [24] and Jannesari et al. [25] employ ResNet-50 and ResNet-152, respectively. The work in Madduri et al. [26] and Cetindag et al. [27] compare employing standard CNNs versus CNNs combined with Local Binary Pattern (LBP) ( for feature extraction). Das et al. [28] introduced a Deep Multiple Instance (DMI) learning-based CNNs framework, representing the slide as a bag of extracted patches; only the bag label is used for training. An image is labeled as benign if all its patches are benign, and similarly, malignant if all its patches are malignant. The study by Sun et al. [29] aims to predict labels of small patches cropped out of a histopathological whole-slide image. Next is a sliding window method to produce a Ductal Carcinoma In Situ (DCIS) probability map. Finally, given the probability map, a tumor border of DCIS is produced and delineated with marching cubes to facilitate pathologists' review and assessment. The work in Qi et al. [13] employs active learning to select unlabeled samples for annotation and a deep learning model to update the increasing training set iteratively. The primary purpose of their work is to alleviate the burden of large-scale annotation for such image classification. The work in El-Agouri et al. [14] used CNNs to predict breast cancer; they collected their private dataset and applied ResNet50 for feature extraction. The work by Gultekin et al. [7] applies a two-tier tissue decomposition method for defining a set of multi-typed objects in an image. These objects are defined by combining texture, shape, and size information, and they may correspond to individual histological tissue components and local tissue subregions of different characteristics. The authors also define a metric called the "dominant blob scale" to characterize the shape and size of an object with a single scalar value. The study by Chen et al. [19] trains SENet deep learning model on histopathology of the liver to classify the different types of liver cancers. The study compared their results with four deep learning models: VGG16, ResNet50, ResNet-CBAM, and SKNet. While the work by Da et al. [15] try to quantify the morphological characteristics and atypia of Signet Ring Cell Carcinoma (SRCC), a malignant tumor of the digestive system, using CNNs. The work by Foroughi et al. [20] studies the ability to understand biological features and interpret them to enhance the prediction accuracy of CNNs. The authors, Boktor et al. [30] propose virtual histological staining to reduce the time needed to prepare histopathological images. The study by Djouima et al. [31] applies Deep Convolution Generative Adversarial Network (DCGAN) to classify breast cancer tumors; for feature extraction, they use DensNet201; they conclude that DCGAN is an efficient prediction for breast cancer image classification.

## 3. Preliminaries

This section provides background information about the techniques and technologies used in this document.

### 3.1. Convolutional Neural Networks (CNNs)

CNNs are deep learning algorithms that process visual data (images). CNNs comprises several layers: input, hidden, and output. CNNs have been used for various tasks, including object recognition, facial recognition, and automatic labeling of images. CNNs tolerates errors or noise in input data. The CNNs have two phases: the feature extraction phase and the learning phase, as shown in Figure 1. The feature extraction phase is where the network extracts features from the input data. In the learning phase, the network learns how to use those features to classify images correctly.

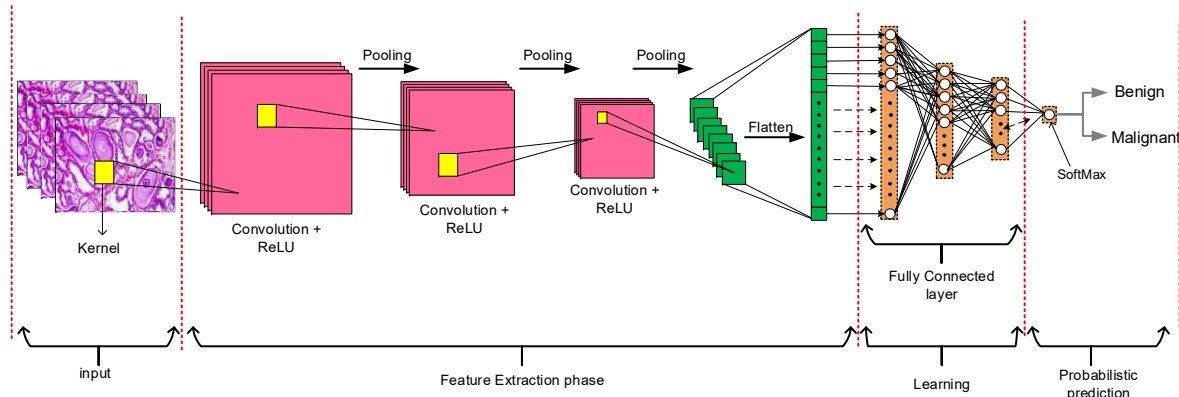

**Figure 1.** Convolutional Neural Networks (CNNs).

The first step in extracting features is to divide the image into small squares, or "kernels". The kernels then scan across the image, and at each location, they calculate a set of values that correspond to essential characteristics of the image. The first layer extracts basic features from the input image, such as edges and corners. The second layer builds on these features to create more complex features such as shapes, edges, and textures. This set of extracted features is then used as input to the learning phase, where it is used to train the model. Once the model is created, it can be used to predict new samples that it has not seen before.

### 3.2. Measurement and Performance Evaluation Methods

There are various measurement techniques available, each with its strengths and weaknesses. The following evaluation metrics are used for this work:

### 3.2.1. Area Under ROC Curve

One of the most important evaluation metrics for assessing the effectiveness of any classification model is the AU-ROC/AUC curve. ROC (Receiver Operating Characteristic) is a probability curve, and AUC (Area Under the Curve) represents the degree or measure of separability. It demonstrates how much the model is capable of differentiating between classes. The higher the AUC, the better the model is at predicting Class 0 as 0 and Class 1 as 1. The classification performance areas are shown in Figure 2.

It is usually computed from the ROC curve, which plots the True Positive Rate (TPR) against the False Positive Rate (FPR) for different cutoff points. The AUC can be interpreted as the probability that a randomly chosen positive example is more likely to be from true positives than false positives.

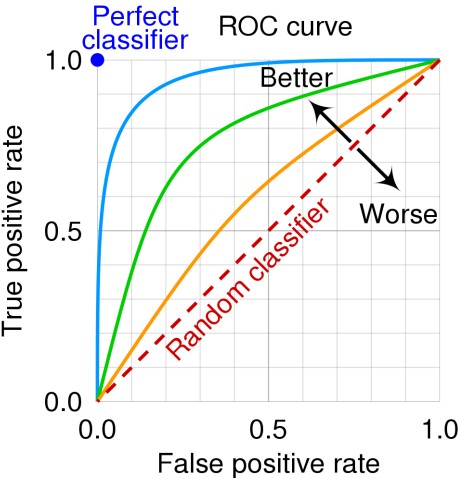

**Figure 2.** Receiver Operating Characteristic (ROC) Curve.

### 3.2.2. Training and Validation Loss

There are two types of losses in ML: training loss and validation loss. The training loss is the amount of error that the algorithm experiences while learning from the data. It provides information on how well our model parameters fit the data. However, it does not tell us how well our model performs on a new data point. The validation loss gauges how close our model comes to correctly predicting labels on a different dataset. In other words, it indicates whether the model has an overfitting or underfitting problem.

### 3.2.3. Precision, Recall, and F1-score Performance Measure

Precision measures how many of the predictions made by the model are correct. The following formula computes the model's precision,

$$Precision = \frac{T_p}{T_p + F_p} \tag{1}$$

In contrast, recall measures how many of all actual target values are correctly predicted by the model. The following formula calculates the recall value of the model,

$$Recall = \frac{T_p}{T_p + F_n} \tag{2}$$

These two metrics are then used to calculate an F1-score, which provides an overall measure of how effective the model is at identifying and predicting target values from a given dataset. The following formula computes the $F$-score value of the model,

$$F\text{-}score = 2\left(\frac{Precision * Recall}{Precision + Recall}\right), \tag{3}$$

where $T_p$ refers to True Positive, $T_n$ refers to True Negative, $F_p$ refers to False Positive, and $F_n$ refers to a False Negative.

### 3.2.4. Confusion Matrix

A confusion matrix is a table that helps to identify the accuracy of a classification algorithm by identifying the number of correct and incorrect classifications. The table consists of two dimensions: actual values and predicted values, as shown in Figure 3. The term "Actually" refers to the correct classifications, while the term "predicted" relates to the value guessed by the algorithm. The numbers in each cell indicate how many times that particular combination occurred.

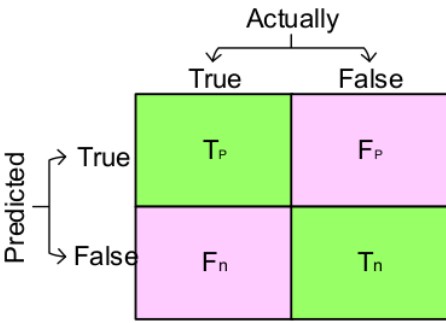

**Figure 3.** Confusion matrix.

## 4. Methodology

### 4.1. Dataset Description

This work uses Breast Cancer Histopathological Image Classification (BreakHis), a publicly available dataset at [21]. As shown in Table 1, the dataset is composed of 7909 microscopic images of breast tumor tissue collected from 82 patients using different magnifying factors: 40x, 100x, 200x and 400x. The dataset contains 2480 benign and 5429 malignant; it has two image sizes: (700 × 460) or (700 × 456) pixels; images are 3-channels, i.e., RGB and 8-bit depth in each channel, using PNG format. The BreaKHis dataset is divided into two main groups: benign and malignant tumors, where benign refers to noncancerous lesions, and malignant refers to cancer lesions. The BreakHis dataset was collected using the Suical Open Biopsy (SOB) method.

**Table 1.** Histopathological images dataset from BreakHis.

| Magnification Factor Used | Benign | Malignant | Total |
|:---:|:---:|:---:|:---:|
| 40x | 652 | 1370 | 1995 |
| 100x | 644 | 1437 | 2081 |
| 200x | 623 | 1390 | 2013 |
| 400x | 588 | 1232 | 1820 |
| Total of images | 2480 | 5429 | 7909 |

### 4.2. Dataset Preprocessing

Data preprocessing is a critical part of any ML algorithm. Obtrusive or pointless features must be removed, and the data must be formatted in a way that the algorithm can understand. This study applies the following data operations:

Skew Data

Data skewing happens when the distribution of data points in a dataset is not consistent, meaning that some values are more common than others. Skewing can cause problems when creating prediction models because it can give an inaccurate insight into what is happening. As displayed in Table 1, the BreakHis dataset has 5429 malignant and 2480 benign images, which might indicate a data skewing risk. In order to avoid data skewing, this work uses the whole dataset from benign, that is, 2480 images, and randomly selects 2780 images out of 5429 in the malignant dataset. In total, the used dataset from benign and malignant is 5260 samples divided into training, validating, and testing, with the following ratio: 60% for training, 10% for validation, and 30% for testing. The following data preprocessing pipeline steps are applied to the dataset:

### 4.3. Image Scaling

There are two different image sizes in the BreakHis dataset collection, i.e., (700 × 460) or (700 × 456) pixels. This work resizes the images to be the same size, i.e., (700 × 460) pixels.

### 4.4. The Dataset Splitting

The dataset is sampled in the following proportions: 60% for training, 10% for validation, and 30% for testing.

### 4.5. Data Transformation

Data transformation is the process of converting data from one format to another. The following augmentation parameters are used for the training dataset:

- Horizontal flipping = 0.4
- Vertical flipping = 0.4
- Image rotation = 20
- For RGB channel normalization, the following values for mean and standard deviation are used: (0.5, 0.5, 0.5) and (0.5, 0.5, 0.5), respectively.

The following data transformations are used for the testing dataset:

- Image resizing = (700 × 460) pixels
- For image normalization, the following values for mean and standard deviation are used: (0.5, 0.5, 0.5) and (0.5, 0.5, 0.5), respectively.

### 4.6. Feature Extraction Phase

Feature extraction is the extraction of meaningful information from data. The first step in extracting features is to divide the image into small squares, or "kernels". The kernels move across the image, and at each location, they calculate values corresponding to the image's important characteristics. The feature extraction goes through layers; the first layer extracts basic features from the input image, such as edges and corners. The second layer builds on these features to create more complex features such as shapes and textures. This process continues until the final layer recognizes a unique feature in the image. Figure 4 shows the feature configuration settings for each layer used in this work.

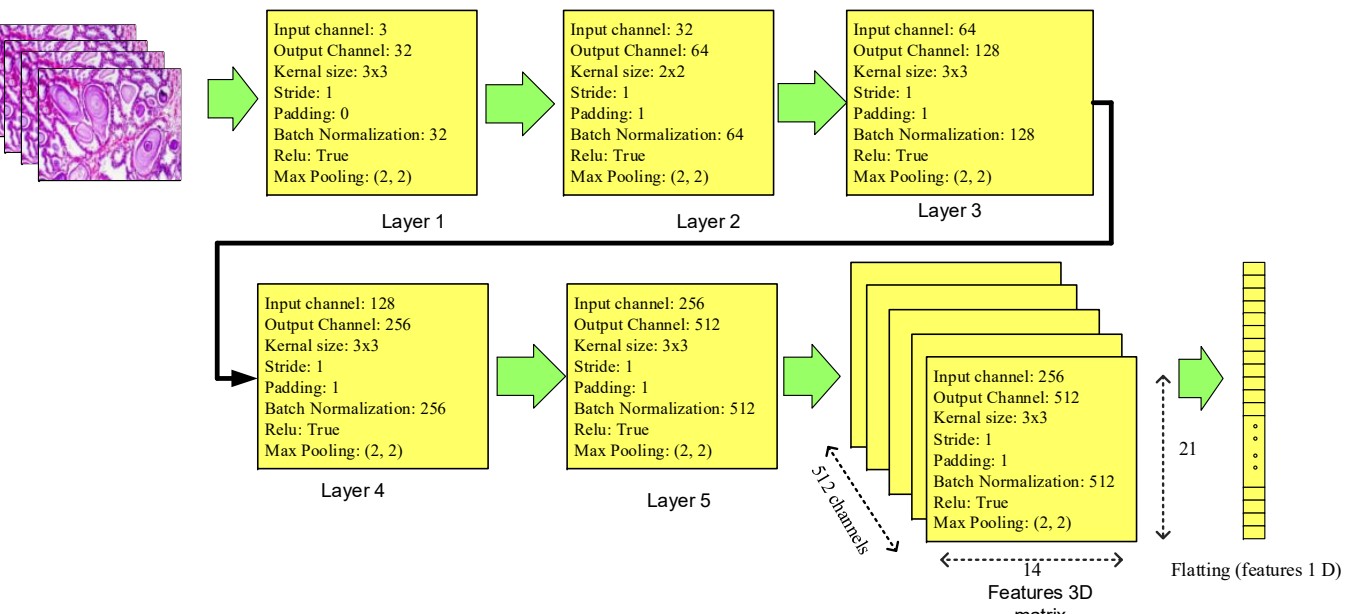

**Figure 4.** Configurations of the feature extraction phase of CNNs.

### 4.7. Classification Phase

As depicted in Figure 1 the classification learning is the second phase of CNNs. The flattened features of the preceding phase are used as input to the learning phase. The ReLU function is the most commonly used activation function in neural network units. For this work, the dropout value is set to 0.4. The last layer of CNNs is a mathematical function called

the softmax, which converts a vector of numbers into a vector of probabilities. The fully connected neural network with the configuration used in this study is shown in Figure 5.

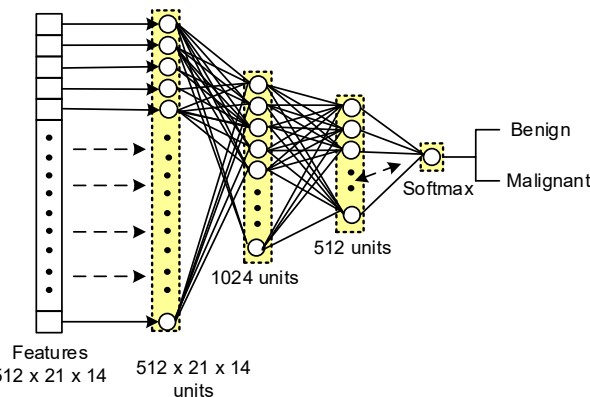

**Figure 5.** Configuration of learning phase.

The model has 156,227,329 trainable parameters. BCELOSs is the loss function that measures the Binary Cross Entropy between the target and input probabilities. The Adam optimization function is selected to apply stochastic optimization. The batch size is 12, and the epoch number equals 15. The model is trained using the GPU.

## 5. Results

### 5.1. AU-ROC/AUC Curve

As illustrated in Section 3.2.1 AU-ROC/AUC measures the ability of a classifier to separate two classes. On the AU-ROC/AUC curve, as shown in Figure 6A, the model scored more than 0.93 out of 1 where 1 being the maximum value.

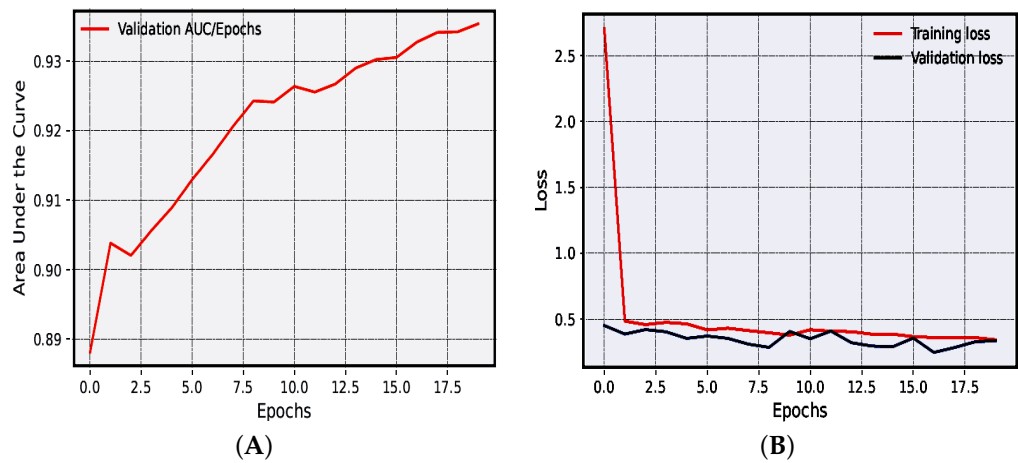

**Figure 6.** Model performance. (**A**) ROC. (**B**) Training and validation loss.

### 5.2. Training Loss and Validation Loss

As described in Section 3.2.2, the training and validation loss function is usually used to evaluate the performance of a model on a hold-out set (validation portion). The created model has low training and validation loss, as shown in Figure 6B. Small training and validation losses indicate that the model has a soft overfitting problem and can be generalized to predict future events (cases) with acceptable accuracy.

As stated earlier, 1736 samples are used for performance evaluation (testing). After applying image preprocessing, the testing sample was randomly selected from the BreakHis dataset. The selected sample contains 792 benign and 944 malignant cases. Using the results from the testing phase, the model was also evaluated using precision, recall, and F1-score

performance metrics, as shown in Table 2. The model scores slightly better at predicting cancerous cases than when predicting normal ones. The calculations of these performance matrices are introduced in Section 3.2.3.

**Table 2.** Normal and cancer prediction accuracy using various classification matrices.

|  | **Precision** | **Recall** | **F1-Score** |
|---|---|---|---|
| Normal | 0.88 | 0.9 | 0.89 |
| Cancer | 0.91 | 0.9 | 0.9 |

Figure 7A depicts the number of times the model predicts correctly and incorrectly. Based on the results shown in Figure 7A, the model has a 90% accuracy level as it correctly predicts 1558 out of 1736 testing samples.

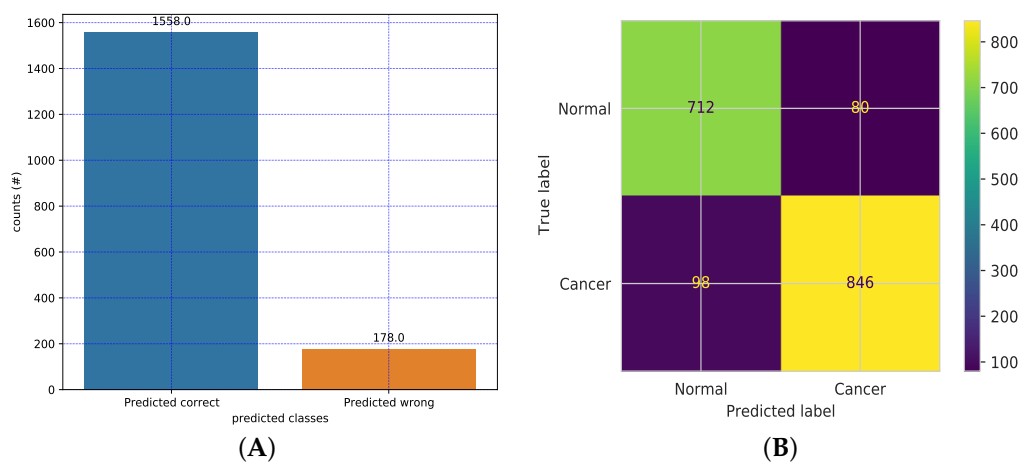

**(A)**          **(B)**

**Figure 7.** Overall prediction accuracy. (**A**) Cancerous class prediction. (**B**) Confusion matrix.

The confusion matrix, depicted in Figure 7B, exhibits greater detail about model performance; it indicates the number of correct and incorrect classifications and the number of times each class was confused with another. From the confusion matrix, 80 normal samples were predicted as cancerous, and 98 cancerous samples were predicted as normal. It also shows that the prediction accuracy percentage, in terms of normal predictions, is $\frac{712}{792} = 0.89$, while, in terms of cancerous predictions, it is $\frac{846}{944} = 0.89$.

As remarked throughout this study, the aim is to study the impact of distinct magnification factors of the histopathological image on model accuracy. The aim is to decide which magnification factor assists the model in achieving the best learning rate. The number of histopathological images for the testing dataset and the combined training and validation dataset is shown in Figure 8A,B. The numbers from Figure 8A,B indicate that no skewing problem is present and that the model is equally exposed to almost comparable numbers of magnification factors during the training and testing phases. The goal at this point is to determine which magnification factor helped the model understand the structure of cancerous or non-cancerous cells more clearly. Which magnification factor, in other words, is characterized as good data, and which one represents noisy data? As accuracy will increase if we only train on the one with more informative features, it is advantageous to use the proper magnification factor that has the most informative features.

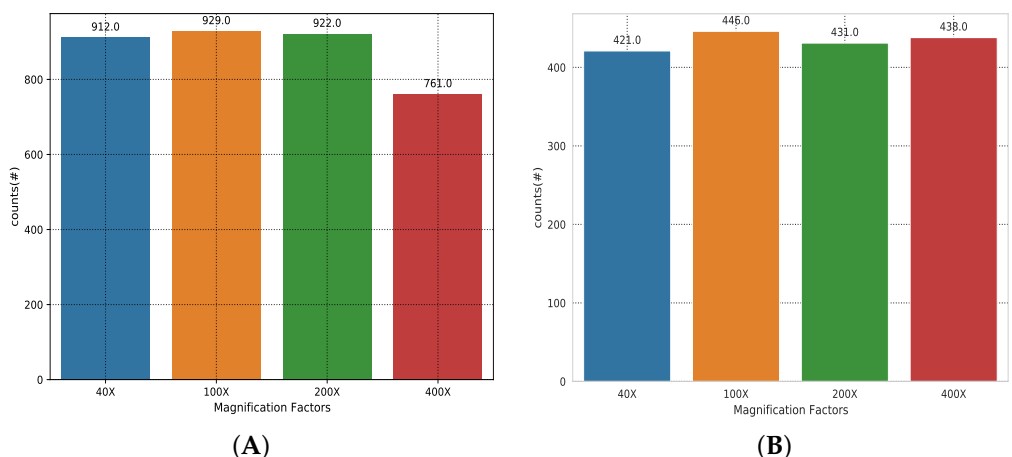

**Figure 8.** Number of magnification factors in the training and testing datesets. (**A**) Training dataset. (**B**) Testing dataset.

After prediction, the testing dataset is separated into four distinct datasets, each representing one magnification factor; one dataset for each magnification factor. The confusion matrices for the four distinct magnification factor datasets are shown in Figure 9. From the shown results, it is clear that, at magnifications of 200x and 400x factors, the model performed better in achieving fewer false-positive and false-negative detections.

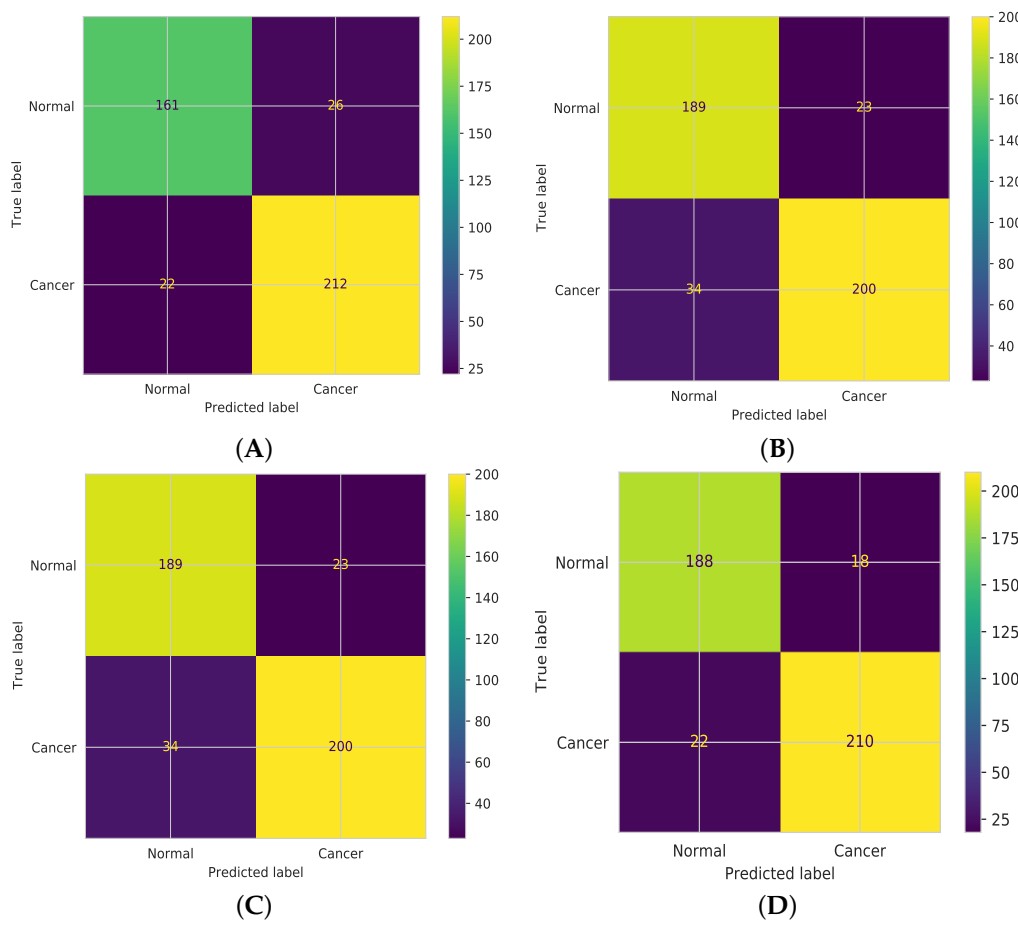

**Figure 9.** Confusion matrix for the four different magnification factors. (**A**) 40x. (**B**) 100x. (**C**) 200x. (**D**) 400x.

Figure 10 shows that the model achieves the highest accuracy when using 200x magnification, followed by a 400x magnification factor. The model performs worse when employing 100x and 40x magnification factors. This conclusion shows that the model learns more effectively from 200x and 400x magnification factors than from 100x and 40x.

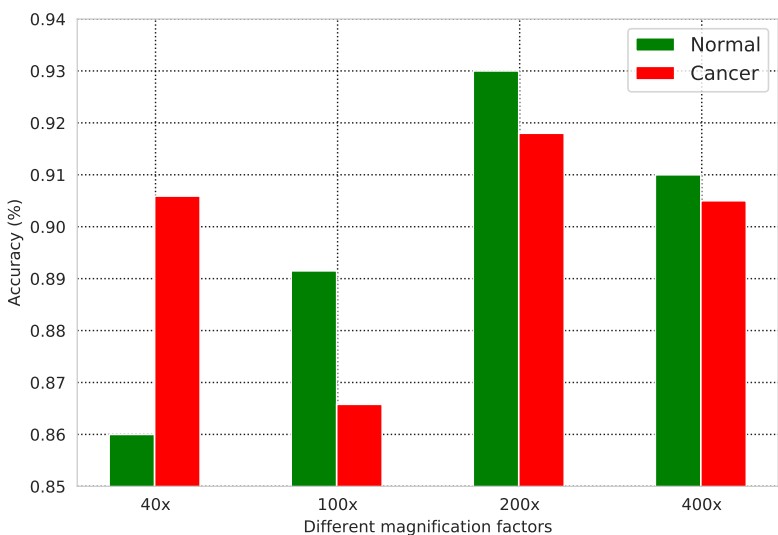

**Figure 10.** Benign or malignant prediction accuracy for the various magnification factors.

Precision, recall, and F1-scare metrics are used to further emphasize the differences in prediction performance among the four magnification factors. The model performs better using 200x and 400x magnification factors, as shown in precision, recall, and F-score measurements, as shown in Figure 11. Noticeably, 200x slightly outperforms 400x in all of the used performance metrics. Furthermore, it is crucial to note that the model performs better at predicting cancerous cells than healthy cells.

Figure 12 displays the weighted average accuracy difference among the various magnification factors. The model achieves high accuracy by using a 200x magnification factor followed by a 400x magnification factor. These findings imply that the model accomplishes a higher learning rate from 200x and 400x magnifications factors.

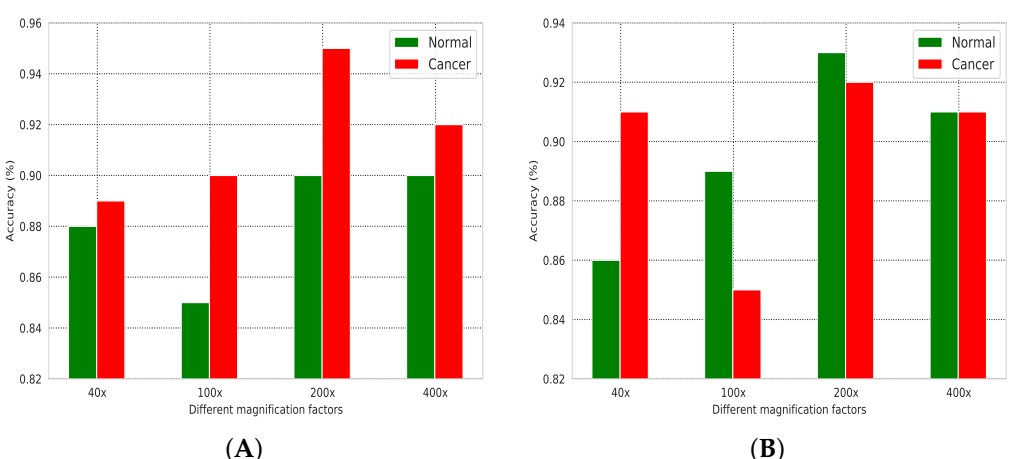

**Figure 11.** *Cont.*

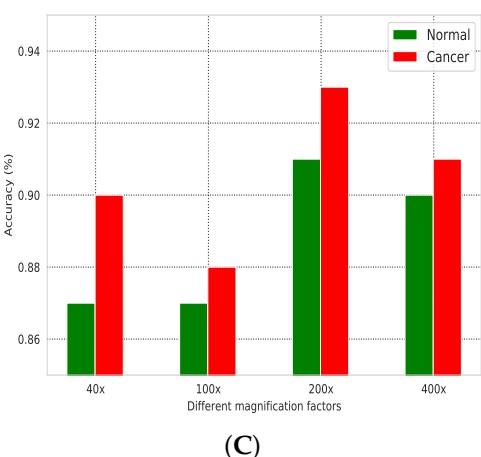

(**C**)

**Figure 11.** Accuracy difference using precision, recall, and F1-score. (**A**) Precision. (**B**) Recall. (**C**) F1-score.

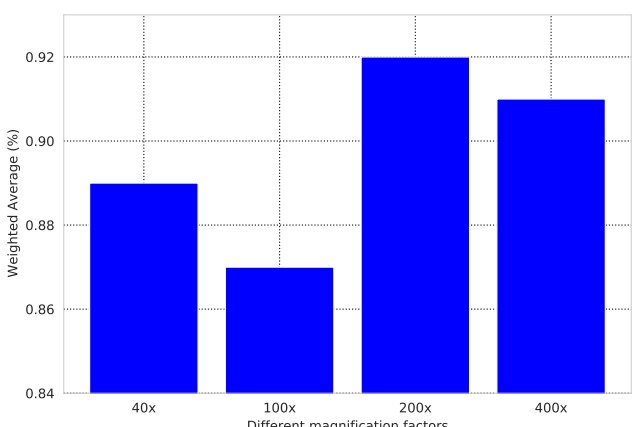

**Figure 12.** Weighted average accuracy difference.

The softmax function at the CNNs' output layer predicts a class (i.e., benign or malignant) with a specified probability value. Table 3, for example, displays the prediction probability values for samples chosen at random from the testing pool. For instance, rows 3, 4, and 6 are predicted correctly as cancerous samples with a high probability, i.e., 0.8, 0.94, and 0.99, respectively. Similarly, the sample in row 9 is correctly predicted but with a probability of 0.53. The samples in rows 3, 4, and 6 have higher prediction probability values than those in row 9. In other words, the model is more certain about the former than the latter.

The model's prediction probabilities at various magnification factors are highlighted (via using Kernel Density Estimator (KDE) ) to assess the model performance difference at various magnification factors. KDE is a method to estimate the Probability Density Function (PDF) from a finite dataset. KDE is a non-parametric method used ( in this study) to display the prediction probability across all magnification factors. KDE is similar to histograms but utilizes other properties such as smoothness or continuity by using the right kernel. Figure 13 shows the KDE of the four magnification factors.

**Table 3.** Prediction probabilities for randomly selected samples from testing dataset.

| # | Histopahological Image Name | T Label | Pred. Label | Pred. Probability |
|---|---|---|---|---|
| 1 | SOB_B_F-14-23060AB-40-011.png | 0 | 0 | 0.064793758 |
| 2 | SOB_B_F-14-9133-200-038.png | 0 | 0 | 0.111493707 |
| 3 | SOB_M_DC-14-13412-100-004.png | 1 | 1 | 0.802534401 |
| 4 | SOB_M_PC-14-15704-200-024.png | 1 | 1 | 0.945152342 |
| 5 | SOB_B_F-14-9133-200-017.png | 0 | 0 | 0.069869727 |
| 6 | SOB_M_DC-14-11520-100-008.png | 1 | 1 | 0.998456001 |
| 7 | SOB_B_TA-14-19854C-200-016.png | 0 | 0 | 0.020913977 |
| 8 | SOB_B_F-14-25197-400-034.png | 0 | 1 | 0.631648719 |
| 9 | SOB_M_DC-14-11031-40-010.png | 1 | 1 | 0.534332812 |
| 10 | SOB_M_PC-14-19440-400-028.png | 1 | 1 | 0.998934567 |

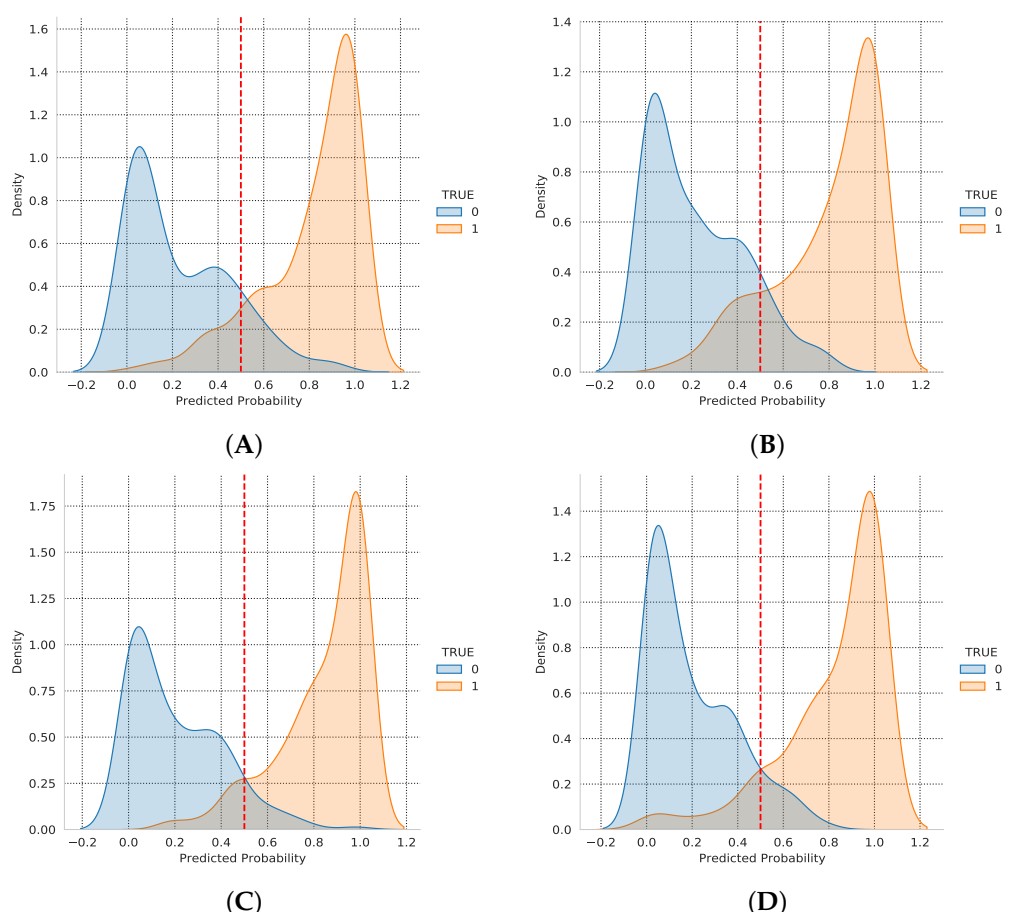

**Figure 13.** KDE of the four magnification factors. (**A**) 40x. (**B**) 100x. (**C**) 200x. (**D**) 400x.

The gray area in the figures represents the area where there is a confusion, benign tumors are mistaken for malignant tumors or vice versa, resulting in inaccurate predictions. Those samples on the left of the border line are predicted non-cancerous cells while they are cancerous; those samples on the right of the border line are denoted as cancerous cells while they are not. Comparing the confusion areas of the different magnifcation factors (displayed in Figure 13A–D, the magnification factor of 200x has a minor area followed by a 400x magnification factor. It is evident that the prediction probability of cancerous cells is higher than those of non-cancerous cells; this is shown in all magnification factors of

Figure 13A–D. Figure 13C for 200x has the highest probability density level, indicating that the model is more confident when learning and predicting from this magnification pool.

The shaded area shrinks as the prediction accuracy increases until it reaches zero when there are no prediction errors. Figure 14A,B show the PDF overlap of the four magnification factors, where Figure 14B is just a zoomed-in view of the confusion area shown in Figure 14A. The shaded area is where the model does not have a very high prediction probability of whether a cell is cancerous or non-cancerous. Figure 14A,B clearly show that 200x and 400x have fewer shaded areas in the confusion area; this indicates that the model achieved a higher learning rate from magnifications of 200x and 400x factors compared with 40x and 100x magnification factors.

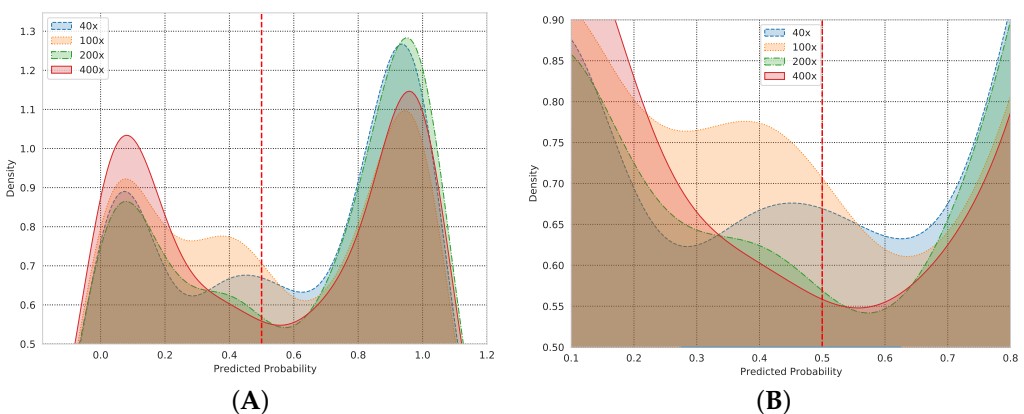

**Figure 14.** Model's confusion area for the four magnification factors. (**A**) Confusion area. (**B**) Confusion area zoomed in.

To prove the stated conclusion reached from Figure 14A,B, the count of samples from all magnification factors is calculated in the confusion area, as illustrated in Figure 15. To perform the counting, the confusion area limits is defined as the range of the prediction probability between 0.3 and 0.7. Figure 15 shows that magnification factors of 200x and 400x have the lowest number of samples in the confusion area. The magnification factor of 200x has fewer sample counts in the confusion area.

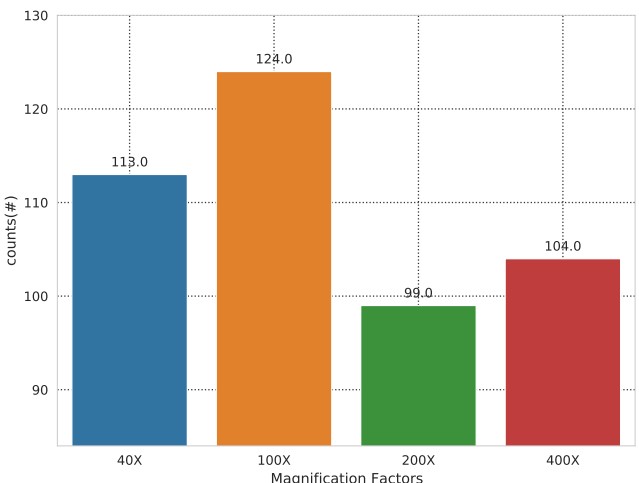

**Figure 15.** Samples count per magnification factor in the confusion area.

## 6. Discussion

The histopathologist uses different magnification factors to capture the unique manifestation of cell deformation. This is because every magnification factor reveals distinctive features of the tissue under investigation. For instance, high-power magnification factors are beneficial for examining tissues in great detail. On the other hand, using low-power magnification helps understand the tissues' overall structure. The disadvantage is that it can be difficult to distinguish between normal and abnormal tissues. Histopathologists usually use different magnification factors to gain information from each level to construct a conclusion. The justification mentioned above holds for human histopathologists, but does it also hold for machine learning algorithms? In particular, do CNNs benefit from training at various magnification levels? The work in this paper studies the impact of the magnification factor on the learning rate. The paper tries to answer: what is the best magnification factor so that the model achieves high prediction accuracy? Table 4 lists the work done in the literature about breast cancer classification using histopathological images. Except for the work in [6,17], the literature did not discuss the effects of various magnification factors on the model's prediction accuracy. The authors in [17] show the accuracy difference among different magnification factors without describing their findings and providing a conclusion. The impact of various magnification factors on learning rate is not the focus of [17] compared to the work done in this paper. A study of the effects of various feature extraction, magnification factors, and classifiers is presented in [6]. The latter work aims not to study which magnification factors achieve a better learning rate; their focus is to evaluate the overall performance as there is no in-depth analysis or conclusion of which magnification factor is the best.

This study thoroughly examines how different magnification factors affect the model's ability to predict classes, i.e., benign versus malignant. The results of this work showed that the model achieved a higher accuracy (learning rate) when using 200x and 400x magnification factors. The latter conclusion is noticeably clear when using the following performance matrices: precision, recall, F-Score, confusion matrix, and weighted average accuracy. The kernel density estimator shows that the model predicts higher probability values when using 200x and 400x magnification factors. This assures that the model finds satisfying features that make the model easy to learn and predict. The model has fewer predictions in the confusion area when learning from 200x and 400x magnification factors, as shown in Figure 15. The conclusion of this study is that it is advisable to use magnification factors of 200x to train and predict. It is worth trying to consider 200x as the center value of the magnification factors and using a magnification factor range (for example, n factor values less and higher than the center); the range is centered around 200x. This will be the future work of this paper.

**Table 4.** Literature review for BC prediction using ML trained on histopathological images.

| Auth. | Dataset | Modality | Methodology | Preprocessing | Objective |
|-------|---------|----------|-------------|---------------|-----------|
| [3] | Kaggle invasive ductal carcinoma | HI | SVM and LR | Feature reduction using PCA and HOG and Canny edge detection | BC classification, i.e., B or M |
| [4] | BreakHis [21] | HI | SVM and LR | CNN (for feature extraction) + SVM (for classification) or CNN (for feature extraction) + LR (for classification) | BC classification, i.e., B or M |
| [5] | Kaggle dataset | HI | SVM and CNN for feature extraction and K-means for tumor extraction. | Segmentation algorithm GA and K-means | locate and extract tumor cells |

**Table 4.** *Cont.*

| Auth. | Dataset | Modality | Methodology | Preprocessing | Objective |
|---|---|---|---|---|---|
| [6] | BreakHis [21] | HI | SVM, CNN, RF, QDA | HOG, WPT, ResNet, and PCA for feature extraction | BC classification, i.e., B or M |
| [7] | locally prepared | HI | SVM | Tissue decomposition to locate multityped objects | image decomposition toward colon cancer classification |
| [8] | Vietnamese Dataset VB-Can | HI | SVM, LR, NB, DT, RF | For feature extraction they use VGG16, ResNet50, GoogleNet | BC classification, i.e., B or M |
| [9] | 430 patients collected from two laboratories | HI + age, sex, lesion site | CNNs | Fusion of patient information with an image | BC classification, i.e., B or M |
| [10] | BreakHis [21] | HI | CNNs | Using original CNN architecture | BC classification, i.e., B or M |
| [11] | BreakHis [21] | HI | CNNs | Feature extraction using DenseNet-201 | BC classification, i.e., B or M |
| [12] | MIAS mammograms | HI | CNN | Feature extraction using the Entropy Function | BC classification, i.e., B or M |
| [13] | BreakHis [21] | HI | Active learning | image labeling using Entropy-based query strategy | BC labeling |
| [16] | Kaggle BC HI | HI | Deep RNNs | Convert RGB image to 7 channels | BC classification, i.e., B or M |
| [17] | BreakHis [21] | HI | RF, SMO | Intensity Histogram Co-occurrence matrix | BC classification, i.e., B or M |
| [18] | —— | HI | Fussy classification | Geometric transformation from RGB to HSV | Cancer Cells classification, i.e., B or M |
| [22] | Hospital of the University of Pennsylvania | HI | DBNs + LR | Applying RoI extraction | BC classification, i.e., B or M |
| [24] | BreakHis [21] | HI | CNN | Feature extraction using ResNet50 | BC classification, i.e., B or M |
| [25] | Tissue Micro Array database and BreakHis [21] | HI | CNN | Feature extraction using ResNet152 | BC classification, i.e., B or M |
| [26] | 100 images from unknown source | HI | Origin CNN or CNN + LBP (feature extraction) | First RGB to gray conversion followed by LBP | BC classification, i.e., B or M |
| [28] | BreakHis [21] | HI | DMI based on CNN | images reduced to bag of labels | BC classification, i.e., B or M |
| [29] | —— | HI | AlexNet + CNN, VGG-11 + CNN, and ResNet-18 + CNN | RoI annotation then extract image patches | predict and localize tumour tissue region |

## 7. Conclusions

This study conducts a thorough analysis to determine which magnification factors, i.e., 40x, 100x, 200x and 400x, are desirable and produce the highest prediction accuracy. This study showed that, in comparison to 40x and 100x, using 200x and 400x magnification factors during training and testing improved the CNNs model's prediction accuracy. More specifically, this study finds that the CNNs model performs better when trained and tested on 200x than it does on 400x. The kernel density estimator demonstrates that the model predicts higher probability values when 200x and 400x magnification factors are used. The results also illustrated that the model has fewer predictions in the confusion area when learning from 200x and 400x magnification factors. The study's findings support the usage of 200x magnification factors for training and prediction.

For the future scope of this study, is considering 200x as the center value of the magnification factors. Then a range of magnification factors, around the magnification center

(200x), is used; for example, using n factor values less and higher than the magnification center to train and test the model.

**Funding:** This research received no external funding.

**Institutional Review Board Statement:** Ethical review and approval were waived for this study as the Dataset is publicly available from the Department of Informatics (DInf) and Graduate Program in Computer Science (PPGInf) of the Federal University of Parana (UFPR).

**Informed Consent Statement:** Not applicable.

**Data Availability Statement:** The dataset used for this study can be found in the following link https://web.inf.ufpr.br/vri/ (accessed on 17 January 2022). The dataset was collected and prepared by the Laboratory of Vision, Robotics and Imaging (VRI). VRI was created in 2010 and it is part of the Department of Informatics (DInf) and Graduate Program in Computer Science (PPGInf) of the Federal University of Parana (UFPR).

**Conflicts of Interest:** The authors declare no conflict of interest.

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
