# Peer review of "Optimal Histopathological Magnification Factors for Deep Learning-Based Breast Cancer Prediction"

_asi, doi:10.3390/asi5050087_

Round 1
Reviewer 1 Report
This research is well organized and the added information is appropriate. We believe that this research adds to the science of knowledge. The authors should address all of the following concerns accurately.
- Paper title requires improvement
- Literature Review Section: It prefers to replace the word "authors" (page3-line114) with the names of researchers (please check the entire paper). Also, it should structure this section and add more references.
- Why is Figure 13 shown before Figure 12?
- English Writing: This article requires minor proofreading to address English writing issues. There are some typos, grammar, and spelling issues. Some paragraphs are long and require breaking into smaller paragraphs. Authors should carefully/accurately check the entire article to remove typos and grammatical mistakes.
- References list: The number of references is not sufficient for this study. The authors must follow the format depending on the MDPI-Applied System Innovation journal style. For instance, references are not arranged in the text. The names of the researchers do not follow the style of the journal. Some papers' names in the references list begin with an uppercase letter in each word such as [3], [4], [5] … etc. and other references use an uppercase only in the first word such as [1], [10], [11] … etc. The double quotation should be removed from the research titles. Authors should fix all problems in the references list and scrutinize them carefully.
Author Response
This research is well organized and the added information is appropriate. We believe that this research adds to the science of knowledge. The authors should address all of the following concerns accurately.
Thank you for the valuable comments; I am sure they improved the quality of this work.
- Paper title requires improvement
The title has been changed to the following,
“Optimal Histopathological Magnification Factors for Deep Learning-based Breast Cancer Prediction”
- Literature Review Section: It prefers to replace the word "authors" (page3-line114) with the names of researchers (please check the entire paper).
Names are added to all references. Please check the paper.
Also, it should structure this section and add more references.
Eight more references, along with their descriptions, are added.
- Why is Figure 13 shown before Figure 12?
It was a mistake in the Latex code. The figure order is fixed now. Thank you for pointing that out.
- English Writing: This article requires minor proofreading to address English writing issues. There are some typos, grammar, and spelling issues. Some paragraphs are long and require breaking into smaller paragraphs. Authors should carefully/accurately check the entire article to remove typos and grammatical mistakes.
I went through the manuscript more than once, and I found a few typos, which I fixed. I break down sentences and paragraphs whenever possible.
References list: The number of references is not sufficient for this study.
More references with their descriptions are added to the paper as I explained above
The authors must follow the format depending on the MDPI-Applied System Innovation journal style. For instance, references are not arranged in the text. The names of the researchers do not follow the style of the journal.
The references are arranged according to the first appearance in the text. However, I double-checked and made some changes now. They are arranged in the order of their appearance in the text.
Some papers' names in the references list begin with an uppercase letter in each word such as [3], [4], [5] … etc. and other references use an uppercase only in the first word such as [1], [10], [11] … etc.
Yes, you are right. Thank you for pointing that out. I fixed them now.
The double quotation should be removed from the research titles. Authors should fix all problems in the references list and scrutinize them carefully.
Double quotas are removed, and I double-checked that the references are consistent.

Reviewer 2 Report
In this manuscript, Dr. Abduladhim Ashtaiwi describe a study of prediction accuracy analysis of deep learning models at different magnification factors of breast cancer histopathologies. Dr. Abduladhim Ashtaiwi found that training CNNs on 200x and 400x magnification factors increased the prediction accuracy compared to training on 40x and 100x. In particular, the CNN model performs better when trained on 200x than on 400x.
The overall subject is meaningful and worthy of study. The methodology is well used and the method used in the analysis has increased the importance of this study. Generally, the results obtained it is good. I feel that it is suitable for publication in this journal but, after the authors should accept the revisions of their paper, particularly on the following points:
1. There are too many figures in the manuscript. It is suggested to combine similar figures into a figure, for example, Figures 6 and 7 into (a) and (b) in one figure. It is best to reduce the number of figures in the whole manuscript within 10.
2. Page12: The order of Figures 12 and 13 should be interchanged in the manuscript.
3. L427-434, page18: These words are not the conclusion of this manuscript. It is suggested to delete them.
Author Response
In this manuscript, Dr. Abduladhim Ashtaiwi describe a study of prediction accuracy analysis of deep learning models at different magnification factors of breast cancer histopathologies. Dr. Abduladhim Ashtaiwi found that training CNNs on 200x and 400x magnification factors increased the prediction accuracy compared to training on 40x and 100x. In particular, the CNN model performs better when trained on 200x than on 400x.
Thank you for your valuable comments. They improve the quality of the paper.
The overall subject is meaningful and worthy of study. The methodology is well used and the method used in the analysis has increased the importance of this study. Generally, the results obtained it is good. I feel that it is suitable for publication in this journal but, after the authors should accept the revisions of their paper, particularly on the following points:
- There are too many figures in the manuscript. It is suggested to combine similar figures into a figure, for example, Figures 6 and 7 into (a) and (b) in one figure.
I combined figures 6 and 7 into one figure. I also combined 8 and 9 into one figure. I checked whether I could combine other figures, but it is not visible as they illustrate content that differs from one section to another.
- It is best to reduce the number of figures in the whole manuscript within 10.
I went through all the figures and tried to delete some of them, but I found that the manuscript would lose some of its clarity and objective in case I removed any. To keep it as clear as possible, I opted to keep the remaining figures.
- Page12: The order of Figures 12 and 13 should be interchanged in the manuscript.
It was a mistake in the Latex code. The figure order is fixed now. Thank you for pointing that out.
- L427-434, page18: These words are not the conclusion of this manuscript. It is suggested to delete them.
True, removing that part of the text makes the reader immediately understand the conclusion.
